# Emerging Roles of the Endoplasmic Reticulum Associated Unfolded Protein Response in Cancer Cell Migration and Invasion

**DOI:** 10.3390/cancers11050631

**Published:** 2019-05-06

**Authors:** Celia Maria Limia, Chloé Sauzay, Hery Urra, Claudio Hetz, Eric Chevet, Tony Avril

**Affiliations:** 1Proteostasis & Cancer Team, Institut National de la Santé Et la Recherche Médicale U1242 Chemistry, Oncogenesis, Stress and Signaling, Université de Rennes, 35042 Rennes, France; cema201089@gmail.com (C.M.L.); Sauzay.Chloe@chu-amiens.fr (C.S.); eric.chevet@inserm.fr (E.C.); 2Centre Eugène Marquis, 35042 Rennes, France; 3Biomedical Neuroscience Institute, University of Chile, 8380453 Santiago, Chile; hery.urra@gmail.com (H.U.); claudio.hetz@gmail.com (C.H.); 4Center for Geroscience, Brain Health and Metabolism (GERO), 8380453 Santiago, Chile; 5Institute of Biomedical Sciences (ICBM), Faculty of Medicine, University of Chile, 8380453 Santiago, Chile; 6The Buck Institute for Research in Aging, Novato, CA 94945, USA; 7Department of Immunology and Infectious Diseases, Harvard School of Public Health, Boston, MA 02115, USA; 8Rennes Brain Cancer Team (REACT), 35042 Rennes, France

**Keywords:** cancer, cell invasion, cell migration, ER stress, IRE1, PERK, ATF6

## Abstract

Endoplasmic reticulum (ER) proteostasis is often altered in tumor cells due to intrinsic (oncogene expression, aneuploidy) and extrinsic (environmental) challenges. ER stress triggers the activation of an adaptive response named the Unfolded Protein Response (UPR), leading to protein translation repression, and to the improvement of ER protein folding and clearance capacity. The UPR is emerging as a key player in malignant transformation and tumor growth, impacting on most hallmarks of cancer. As such, the UPR can influence cancer cells’ migration and invasion properties. In this review, we overview the involvement of the UPR in cancer progression. We discuss its cross-talks with the cell migration and invasion machinery. Specific aspects will be covered including extracellular matrix (ECM) remodeling, modification of cell adhesion, chemo-attraction, epithelial-mesenchymal transition (EMT), modulation of signaling pathways associated with cell mobility, and cytoskeleton remodeling. The therapeutic potential of targeting the UPR to treat cancer will also be considered with specific emphasis in the impact on metastasis and tissue invasion.

## 1. Introduction

Cell migration/invasion is one of the cancer hallmarks that drives cancer progression leading to tumor expansion of the adjacent tissues and/or to tumor dissemination through metastasis. These properties compromise the efficacy of the anti-cancer therapeutic approaches such as surgery or irradiation that rely on the existence of defined and limited zones within the tumor site. For instance, in glioblastoma (GBM), the diffuse infiltration of tumor cells into the cerebral neighboring parenchyma renders the complete and safe tumor resection as almost impossible. In turn, this leads to the recurrence of GBM [1,2]. Metastases observed in many tumor types, in concert with anti-cancer drugs tumor resistance, also largely contribute to most of the curative failures in cancers and in cancer-related mortality [3,4]. Therefore, in recent years, targeting tumor invasion/migration has become an attractive approach for the development of new types of anti-cancer treatments [2,5].

Cancer cells can adapt to restrictive microenvironmental conditions associated with nutrient and oxygen deprivation. This occurs by triggering a series of adaptive stress responses that include the Unfolded Protein Response (UPR). The activation of the UPR allows tumor cells to restore Endoplasmic Reticulum (ER) proteostasis [6,7,8]. In the past couple of years, increasing evidence has linked the UPR to cancer progression due to the ability to regulate many cancer cell functions. However, the link between the UPR and the ability of cancer cells to migrate and invade has not been addressed in depth, and only a few examples have been described so far. In this review, after a brief overview of ER stress signaling, we describe the cellular and molecular aspects of the cell migration and their relevance to cancer cell migration/invasion, mainly focusing on brain and skin tumors, and how this can be connected to UPR signaling. Moreover, we discuss the therapeutic perspectives targeting the UPR/cancer cell migration/invasion links to limit the tumor dissemination.

## 2. Mechanisms and Molecular Actors of Tumor Cell Migration and Invasion

Increased tumor cell migration/invasion capacity is one of the hallmarks of cancer [9,10] and leads to tumor dissemination and aggressiveness [11]. To spread within the tissues, tumor cells use migration mechanisms that are similar to those occurring in physiological processes such as embryonic development (Figure 1) [12]. Tumor cell invasion/migration involves diverse patterns of interconvertible strategies including mesenchymal, amoeboid single migration or collective movements (Figure 2).

### 2.1. Different Steps of the Migration Process at the Cellular and Molecular Levels

#### 2.1.1. Polarization of the Migrating Cell

Cell migration can be described as a five-step process [11,12]. In the first step, the moving cell becomes polarized and elongated due to the generation of a protrusion at the leading edge (Figure 1, (1)). These protrusions, composed of parallel and crosslinked actin filaments, take several forms such as broad lamellipodia, or spike-like filopodia [11,13]. This step is initiated spontaneously or by different stimuli including chemokines and growth factors leading to the activation of RHO family GTPases such as RAC1, RHOA and CDC42 [11,12]. RAC1 is a key regulator of migration and localizes to the leading edge of moving cells [14,15]; together with CDC42, these GTPases are involved in the formation of filopodia and lamellipodia whereas RHOA is involved in the formation of stress fibers [16].

#### 2.1.2. Dynamic Interactions of the Migrating Cell with ECM

In the second step, the elongated protrusions form focal contacts with adjacent extracellular matrix (ECM) components which then mature into focal adhesions (Figure 1, (2)). These stable cell-ECM interactions comprise adhesion molecules, most notably transmembrane receptors of the integrin family, and other receptors such as CD44 and syndecans [17]. Intracellularly, focal adhesions are composed by integrin clusters that recruit signaling proteins such as the non-receptor protein tyrosine kinase named focal adhesion kinase (FAK, also known as PTK2 for protein tyrosine kinase 2) that acts a major player in the positive regulation of cell migration. Upon integrin-mediated cell adhesion formation, FAK becomes auto-phosphorylated on Y397 residue leading to its association with SRC and resulting in activation of both kinases. Through carboxy-terminal proline-rich regions, FAK binds CAS (for Crk-associated substrate), a scaffold molecule important for regulating cell migration; and paxillin (PXN), another important molecule for cell spreading and migration events, resulting into more stable and mature focal adhesions [18,19,20]. In a third step, specific surface proteins are recruited near substrate attachment sites leading to the cleavage of ECM components, such as collagen, fibronectin and laminins (Figure 1, (3)). These proteins involved in ECM degradation comprise secreted metalloproteinases such as MT1-MMP, MMP1 and MMP2 (Figure 1, (3)) [11,21]. In this step, one of the major driver of tumor cell invasion though ECM is the invadopodium. Invadopodia are β-actin rich invasive protrusions that degrade the cross-linked networks of EMC which restrict tumor cell motility to cross the epithelia and endothelia cell layers. The capacity of these structures to degrade ECM is attributed to the presence of membrane-bound MMPs such as MT1-MMP and to the release of others MMPs, like MMP2 and MMP9 [22,23].

#### 2.1.3. Cell Contraction and Detachment to ECM Allowing Cell Movement

In the fourth step, intracellular myosin II binds to actin filaments to form actomyosin, thus allowing cell contraction due to a reorganization of the actin cytoskeleton (Figure 1, (4)). The calcium/calmodulin dependent enzyme myosin light chain kinase (MLCK) phosphorylates myosin light chains (MLC) to activate myosin II and generate actomyosin contraction [24]. Dephosphorylation of the myosin light chain by the MLC phosphatase (MLCP) results in myosin II inactivation. RHOA regulates actomyosin contraction predominantly through its effector, ROCK (for RHO-associated protein kinase), which phosphorylates and inhibits MLCP [25,26]. In the last, the detachment of the tail occurs through different actin binding proteins cause actin filament strand breakage thereby promoting filament turnover. FAK causes focal contact disassembly once phosphorylated by AKT1 [27]. Following this focal contact disassembly, integrins detach from the substrate and become internalized through endocytosis for further recycling towards the leading edge (Figure 1, (5)) [11]. Integrin endocytosis is mediated by clathrin, and its adaptor molecules ARH (for autosomal recessive hypercholesteremia) and DAG2 (for Disabled-2) [28]. All these steps are sequential and polarized through the cell in order to induce a positive force that allow cell to move in a specific direction.

### 2.2. Different Cellular Patterns of Migration

#### 2.2.1. Collective Migration

Collective cell migration is a fundamental process that enables the coordinated movement of groups of cells that remain connected via cell-cell junctions (Figure 2, (1)) [29]. In this migration mode, cells remain physically and functionally connected, preserving the integrity of cell-cell junctions during movement. The connected cells require multicellular polarity and “supra-cellular” organization of the actin cytoskeleton generating traction and protrusion force for migration and maintaining cell-cell junctions (Figure 2, (1)). Migrating groups of cells structurally modify the tissue along the migration path, which may result in the deposition of a basement membrane. All of these processes are guided by chemical and physical cues including chemokines, such as stromal cell-derived factor 1 (SDF1; also known as CXCL12), and members of the fibroblast growth factor (FGF) and transforming growth factor-β (TGFβ) families [30]. Cell-cell adhesion is mediated by adherens junction proteins, including cadherins, other immunoglobulin superfamily members and integrins, all of which directly or indirectly connect intracellularly to the actin and/or intermediate filament cytoskeleton. Several mechanisms polarize the cell group into “leader” cells that guide “follower” cells. Leader cells localize at the front of a moving group, where they receive guidance signals and instruct follower cells, at the rear, into directional migration through chemical and/or mechanical signaling [31]. This polarity of a group of cells is probably cell type and tissue/context specific and might result from the differential expression of surface receptors, extracellular inputs and downstream intracellular signals that define and maintain leader cells. The key components that induce and regulate collective migration include the chemokine receptors CXCR4 and CCR7, mitogen-activated protein kinase (MAPK)/extracellular signal-regulated kinase (ERK), focal adhesion kinase FAK, phosphoinositide-3-kinase (PI3K)/AKT, SRC kinases, NOTCH and RHO GTPases [29,30,32]. The key point of a collective migration is that moving cells influence each other bidirectionally between leading cells and follower cells. Follower cells can also influence the behavior of the leaders to modulate the collective movement. Follower cells can also engage in cell-substrate traction forces, and transmit forces across a longer distance and multiple cell bodies within moving cell sheets [29].

#### 2.2.2. Single Cell Migration/Invasion through a Mesenchymal Mode

Mesenchymal cells move via the five-step migration cycle presented above. Mesenchymal migration can be compared to fibroblast-like motility. Apart from fibroblasts, keratinocytes and endothelial cells, some tumor cells also use this mode of migration. In mesenchymal migration, cells adopt an elongated, spindle-like shape and exert traction on their substrates via focal adhesions associated with actin rich protrusions at the leading edge, such as lamellipodia or filopodia (Figure 2, (2)) [11,33], where RHO-family small GTPases RAC1 and CDC42 are key players [11,33,34].

RAC1 is activated by DOCK3 (for Dedicated Of Cytokinesis 3) and its adaptor molecule NEDD9 to drive mesenchymal migration [34]. Active RAC1 negatively regulates RHO/ROCK signaling, inhibiting cell rounding and promoting mesenchymal movement. The mesenchymal mode requires ECM proteolysis to allow RAC1-dependent actin protrusions to pass through the ECM mesh [34]. Integrins, membrane-type MMPs and other proteases co-localize at fiber binding sites to contribute to ECM degradation. The activation of MMPs and uPA (for urokinase-type plasminogen activator) is required for maintenance of the phenotype and mesenchymal migration [11].

#### 2.2.3. Single Cell Migration/Invasion through an Amoeboid Mode

Amoeboid-like movement has been described in leukocytes and certain types of tumor cells. In amoeboid migration, cells adopt round or irregular shapes (Figure 2, (3)). During locomotion these cells constantly change shape by rapidly protruding and retracting extensions, which allow them to squeeze through gaps in the ECM [33,35]. These cells are highly deformable, due to their lack of focal contacts, allowing them to move at 10- to 30-fold higher velocities than cells that use mesenchymal migration mechanisms [11]. Amoeboid migration involves a range of different sub-modes, such as bleb-based migration or gliding. Bleb expansion is driven by hydrostatic pressure generated in the cytoplasm by the contractile actomyosin cortex [36]. In contrast to the mesenchymal mode of invasion, in the amoeboid-like migration, the contractile polarized actomyosin cytoskeleton is crucial for the generation of the motive force, promoted by the RHO/ROCK signaling pathway [37]. This underlines a mutual antagonistic role between RHO-family GTPases RAC1 and RHO/ROCK that display exclusive functions in mesenchymal and amoeboid-like migration respectively [38,39,40]. By modulating these RHO GTPases, several molecules as such FILGAPP and ARHGAP22 control the mesenchymal/amoeboid switch in cancer cells [34,40,41]. Importantly, the key marker of the amoeboid invasiveness is its independence from ECM proteolytic degradation [35,36].

### 2.3. Migration Strategies Used by Tumor Cells at the Cellular and Molecular Levels

The ability of cancer cells to invade adjacent tissues or to form distant metastases is one of the most life-threatening aspects of cancer. Most solid cancers progress to disseminated metastatic disease, leading to secondary tumors arising in sites distal to the primary tumor [42]. The local invasion of tumor cells to tissues adjacent to the primary tumor site is one of the early steps in the metastatic process and one of the key determinants of the metastatic potential of tumor cells. In order to overcome the ECM barriers and surrounding cells, tumor cells develop abilities to use and switch between different migration modes, i.e., collective, mesenchymal and amoeboid (see above) resulting in the high plasticity of cancer cells. The conversion from epithelial cells to motile individually migrating cells is an intensively studied phenomenon known as epithelial-mesenchymal transition (EMT) [42]. The amoeboid and mesenchymal types of migration/invasion are mutually interchangeable in processes named mesenchymal-to-amoeboid transition (MAT) or amoeboid-to-mesenchymal transition (AMT), respectively (Figure 2) [33]. In the following sections, we describe the tumor migration processes by mainly focusing on two well characterized tumor types (i.e., melanoma and GBM) known to exhibit high migration/invasion properties through a metastatic or a tissue infiltration process.

#### 2.3.1. The Metastatic Process in Melanoma

Cutaneous melanoma is a skin cancer derived from melanocytes is considered as one of the most aggressive malignancies with one of the worst prognoses. The aggressiveness of this malignancy is due to the complexity and dissemination potential of this disease. Metastatic melanoma accounts for approximately 80% of skin cancer-related deaths [43]. Malignant melanoma represents a very relevant model for studying tumor invasion because of its highly metastatic behavior. Melanoma progression involved radial and vertical growth phases. The radial growth phase (RGP) represents early-stage disease and encompasses horizontal growth within the epidermis. When the tumor lesion enters the vertical growth phase (VGP), the repertoire of adhesion molecules changes, the tumor enters the dermis and acquires the capacity to metastasize [44]. At the cellular level, melanoma cells invasion results from a combination of several mechanisms: (i) the epithelial-to-mesenchymal transition, (ii) the loss of cell-to-cell adhesion, (iii) the loss of cell-matrix adhesion, (iv) matrix degradation, (v) chemo-attraction/repulsion and (vi) migration. At the molecular level, all these cellular events are closely regulated and tightly interconnected through the activation of select signaling pathways including MAPK, PI3K and WNT/β-catenin [43,45]. Somatic mutations in the *BRAF* or *NRAS* oncogenes are present in the majority of melanoma cells and lead to the spontaneous activation of the MAPK pathway, promoting cell proliferation, migration and survival [46]. One of the best described phenomena of cell-cell interactions responsible for melanoma progression is the “cadherin switch” [47] by replacing E-cadherin to N-cadherin. This switch is mainly regulated by the PI3K/PTEN pathway through the transcription factors TWIST and SNAI1, two major players of EMT [48]. Loss of E-cadherin may affect the β-catenin/WNT signaling pathway, resulting in upregulation of genes involved in growth and metastasis [44]. Moreover, in malignant melanoma, α4/β1 and αv/β3 integrins play a major role in metastasis dissemination. Indeed the expression of integrin α4/β1 correlates with the development of metastases and is negatively associated with disease-free and overall survival [49]. Moreover, the αv/β3 integrin is highly expressed during the transition from RGP to VGP, suggesting a specific role in melanoma invasion. Indeed, the silencing of integrin αv/β3 in B16 melanoma cells reduces their migratory capacity in vitro and metastatic potential in vivo [50]. Other important players involved in melanoma invasion are metalloproteinases. Protein and activation levels of MMP1, 2, 9 and 13 are upregulated in malignant melanoma [51]. As such, MMP2 cleaves fibronectin into small fragments to enhance the adhesion and migration of human melanoma cells mediated by αv/β3 integrin [52]. In addition to mesenchymal movement, melanoma cells can also adopt amoeboid motility through specific effectors of RHOA, namely ROCK and MLC2 [43], stimulated by the TGFβ/SMAD pathway [53]. RAC1 is involved in mesenchymal migration of melanoma cells, through the adaptor protein NEDD9. *NEDD9* gene is amplified in approximately 50% of melanomas [54]. NEDD9 is a member of the CAS family of proteins that interacts with the guanine nucleotide exchange factor DOCK3 to promote RAC1 activation [55]. Besides, NEDD9 overexpression leads to increased phosphorylation of β3-integrin on Tyr785 in the cytoplasmic domain promoting the assembly of a signaling complex containing β3-integrin, SRC, FAK and NEDD9. Altogether, this leads to an increased activation of RAC1, SRC and FAK and a decreased ROCK signaling that drive an elongated, mesenchymal type invasion [54]. Malignant melanoma represents a very relevant model for studying tumor invasion because of its highly metastatic behavior.

#### 2.3.2. Tumor Migration in Glioblastoma

If most solid tumors spread by metastasis like melanoma, there are exceptions such as glioblastoma (GBM) which is characterized by a diffuse invasion of tumor cells within the surrounding brain parenchyma (referred to as diffuse infiltration hereafter). GBM is the most common primary malignant brain tumors. Despite the aggressive standard of care currently used, including surgery, chemo- and radiotherapy, the prognosis remains very poor. One of the central hallmarks of GBM is the diffuse infiltration of tumor cells throughout the neighboring normal tissues, rendering complete and safe resection almost impossible [56]. GBM cells mainly appear to invade the surrounding brain parenchyma using the mesenchymal form of motility in vivo, in contrast, amoeboid invasion of GBM cells has been only described in vitro [56,57,58]. GBM cells move along myelinated axon tracks and disseminate into healthy brain regions along the vascular basement membrane and the glia limitans externa where fibrous proteins such as collagens, fibronectin, laminins and vitronectin are expressed [56]. GBM cells secrete ECM proteins into the microenvironment and release MMPs for ECM remodeling and to promote their own infiltration. In GBM, matrix metalloproteinases are particularly involved in aggressive tumor cell infiltration [59]. MMP2, MT1-MMP and MT2-MMP activities are highly increased in GBM tumors compared to normal [60,61,62]. MMP2 expression levels correlate with malignant progression in vivo [60,63]. Concomitant with the upregulation of pro-migratory ECM proteins, elevated expression cell adhesion molecules such as integrins receptors and ICAM1 (for intercellular adhesion molecule) has been detected in GBM samples. Integrin receptors reported to be upregulated on glioma cells include α2β1, α5β1, α6β1 and αvβ3. ICAM1 and LFA3 (for lymphocyte function-associated antigen 3) were distinctive markers of GBM [2,64]. A recent study showed that β1 and αv integrins represent the primary adhesion systems for glioma cell migration in different migration models [65]. Interestingly, SRC, FYN, and c-YES kinases belonging to the SRC-family kinase (SFK) are involved in glioma proliferation and motility in vitro [66]. Conversely, LYN, another kinase of this family, shows anti-tumor effect in a glioma orthotopic xenograft model [66]. Components of the FAK/SRC tyrosine kinase migration signaling network are upregulated and activated in GBM suggesting a role of this pathway in tumor invasion [67,68]. The IL6/STAT3 signaling axis is also involved in GBM cell migration by modulating the expression of metalloproteinases MMP2 and MMP9 [69,70], as well as GRP1 (for Glioma Pathogenesis-Related 1) that contributes to GBM stem cell migration [71]. STAT3 is also implicated in mesenchymal GBM progression by modulating the mucin-type protein podoplanin and the EMT-related transcription factors SNAIL and TWIST [72]. Recently we showed that CD90 (THY1) expression controls tumor cell migration/adhesion mainly through SRC signaling. In addition, we show that CD90 expression regulates tumor invasive characteristics in a mouse model and in human tumors [73]. CD90 is a glycophosphatidylinositol anchored glycoprotein considered as a marker for mesenchymal stromal/stem cells that has been earlier described in glioma/GBM specimens and immortalized glioma/GBM cell lines [73].

In the past few years, pharmacological approaches aiming at dampening GBM invasiveness showed promising results in vitro, in mouse models or in clinical trials and targeted metalloproteinases [74,75,76], integrins [77], cytoskeleton reorganization [78], or signaling molecules such as FAK [79,80,81] and SRC [82,83,84,85]. Despite intensive efforts, there has been little improvement in the ability to treat GBM. Hence, understanding cell migration is a necessary first step in developing new “anti-migration” therapies.

## 3. Brief Overview of the Unfolded Protein Response and of the ER Stress Sensors

Cells depend on the production of membrane and secretory proteins to maintain their survival. The production of these proteins is ensured in part by the early secretory pathway comprising the ER and the Golgi apparatus [8]. To adapt to the cellular demand and to the challenges imposed by the surrounding environment, the secretory pathway needs to adjust the associated molecular network involved in protein biogenesis including chaperones, foldases, glycosylation enzymes, oxidoreductases and molecules involved in protein quality control [8]. Despite this elaborate system, a proportion of newly synthesized proteins does not reach the quality criteria and is targeted to the ER Associated Degradation (ERAD) system. When the protein folding demand outweighs the ER folding capacity, improperly folded proteins accumulate in the ER lumen resulting in a condition named ER stress. To combat ER stress and return to a homeostatic situation, an adaptive cellular stress response named UPR is triggered [6,86]. The UPR is exacerbated in the course of tumor development, when cancer cells are exposed to intrinsic and extrinsic challenges, i.e., activation of their oncogenic program or nutrient and oxygen deprivation [6,7].

### 3.1. UPR Signaling Pathways

The UPR activation is controlled by three ER resident transmembrane proteins that act as molecular ER stress sensors: the activating transcription factor 6 alpha (ATF6α, referred to as ATF6), the inositol-requiring enzyme 1 alpha (IRE1α, referred to as IRE1 hereafter) and the protein kinase RNA-like ER kinase (PERK) [6,7,8,87]. Herein, we will provide a brief overview of the UPR signaling pathways and invite the readers to refer to the following reviews on UPR for more details [6,7,8,86,87,88,89,90,91,92,93,94,95,96].

#### 3.1.1. Activation Mechanisms of the ER Stress Sensors

The current dogma reports that activation of these three sensors is regulated by the ER resident chaperone GRP78/BiP. Under basal conditions, GRP78 constitutively associates with the luminal domains of the 3 sensors, thus repressing their activation [7,8,97]. The fine tuning of the ER stress sensors activation has been more precisely described by different mechanisms but a consensus about IRE1 and PERK activation is missing. For instance, several key players have been described such as the importance of the ATPase domain of GRP78 for the interaction with IRE1 [98]; and the involvement of other molecular partners such as the co-chaperone ERDJ4 and the protein disulfide isomerase PDIA6 that participate in the stabilization of the GRP78 and IRE1 interaction [99,100]; and also facilitates IRE1 dimerization during ER stress induced after disruption of ER calcium homeostasis [100]; the heat shock protein HSP47 that favors GRP78 release and stabilizes IRE1 dimerization [101]; or misfolded proteins that can directly interact with IRE1 to trigger IRE1 dimerization/oligomerization [102]. Upon accumulation of misfolded proteins in the ER lumen, GRP78 is released from the ER stress sensors which leads to their activation by allowing IRE1 and PERK dimerization/oligomerization and ATF6 export to the Golgi apparatus [87,97]. ATF6, IRE1 and PERK then trigger downstream signaling pathways to reprogram cells to cope with the stress or to die if the stress cannot be resolved.

#### 3.1.2. ATF6

ATF6 is an ER-localized protein that exists in two isoforms α and ß forming homo- and heterodimers [103,104]. Compared to the isoform ATF6ß, ATF6α appears to be a very potent transcription factor [105]. ATF6 is considered as a natively instable protein [106]. Under ER stress, GRP78 dissociation and disulfide bond modification mediated by the protein disulfide isomerase PDIA5 [107,108] stabilize ATF6 and promotes its export to the Golgi apparatus. In the Golgi apparatus, ATF6 is activated by its cleavage mediated by the S1P and S2P proteases [109,110,111]. This releases an active membrane-free transcription factor, ATF6f, that translocates to the nucleus and induces the transcription of genes mainly involved in protein folding and ERAD such as calreticulin, GRP78, HERPUD1 and SEL1L [7,89,112,113].

#### 3.1.3. IRE1

The cytoplasmic region of IRE1 is composed of two domains with distinct enzymatic functions including a serine/threonine kinase and an endoribonuclease (RNase). During ER stress, IRE1 dimerization/oligomerization leads its trans-autophosphorylation that prompts a conformation change, resulting in the activation of its RNase domain [7]. This RNase has the unique ability to proceed with the excision of 26 nucleotides of a short intronic region of the mRNA of the X-box binding protein XBP1. Together with the tRNA ligase RTCB, IRE1 catalyzes the unconventional splicing of XBP1 to generate a new mRNA with a novel open-reading frame encoding for the transcription factor XBP1s. XBP1s activates expression of genes involved in protein folding, secretion, ERAD and lipid synthesis [7]. When ER stress cannot be resolved, IRE1 RNase also catalyzes the degradation of ER localized mRNA, ribosomal RNA and microRNAs through a process called Regulated IRE1 Dependent Decay of RNA (RIDD), participating to the attenuation of the global mRNA translation. The dual IRE1 RNase activity (XBP1 splicing vs. RIDD) is dependent on IRE1 dimerization/oligomerization state with still debated models [7,86,87]. On the other hand, once IRE1 is activated, IRE1 kinase domain interacts with the adaptor protein TRAF2, and triggers a phosphorylation cascade leading to c-Jun N-terminal protein kinase (JNK) and NFκB activation [114,115]. Upon sustained ER stress, IRE1 activation favors the activation of cell apoptosis through terminal RIDD that cleaves mRNAs non-specifically.

#### 3.1.4. PERK

Upon ER stress, PERK trans-autophosphorylation leads to its activation and phosphorylation of the eukaryotic translation initiation factor 2 alpha (eIF2α) and the transcription factor NRF2 [7,8,92,95]. Phosphorylation of eIF2α leads to the attenuation of the global translation, reducing the folding demand on the ER [7,89,116,117]. Phosphorylation of eIF2α also prompts the translation of the transcription factor ATF4 through a uORF-dependent mechanism [118,119]. Phosphorylation of cytoplasmic NRF2 leads to its dissociation from KEAP1 and its nuclear import [120]. The transcription factors ATF4 and NRF2 induce expression of genes involved in protein folding (via HSF1 that regulates HSP genes), amino-acid metabolism (PHGDH, PSAT1, SHMT2 and SLC genes), antioxidant response (NQO1), autophagy (ATG genes) and apoptosis (CHOP) [7,8,121,122]. Translation restoration is induced by eIF2α dephosphorylation which is catalyzed by the GADD34/PP1c complex [123]. GADD34 is activated downstream of CHOP and its expression results in a negative feedback loop for PERK signaling pathway. If the stress of the ER is maintained, sustained ATF4/CHOP activation leads to the apoptosis of the cell through induction of pro-apoptotic genes related to BCL2 family such as BIM and PUMA [124,125].

### 3.2. Roles of the ER Stress Sensors in Cancer

During tumor development, cancer cells are exposed to several challenges such as acute demands of protein synthesis due to oncogene expression and drastic extracellular conditions linked to hypoxia or low nutrient availability. These challenges require efficient UPR allowing the cancer cells to cope and adapt [88,92,95]. Thus, fingerprints of UPR activation have been found in several types of primary and metastatic tumors including brain, breast, colon, liver, lung, hepatocellular carcinoma and skin (reviewed in reference [126]). For instance, GRP78 and ATF6 mRNAs are up-regulated as the histological grade increases in hepatocellular carcinoma tissues [127]. In human breast carcinomas, high GRP78 and XBP1 protein levels are found in comparison with normal tissue [128]. High GRP78 expression is associated with metastasis and poor prognosis in breast, colon, esophageal, lung and skin cancers [129,130,131,132,133]. Moreover, ablation of the UPR sensors leads to a significant reduction in tumor growth in different types of cancers like colon, pancreatic, breast cancer and GBM [134,135,136,137]. In addition to restoring ER proteostasis, a number of studies have demonstrated that tumor cells hijack the UPR machinery to provide new molecular pathways for supporting tumor development and aggressiveness associated with microenvironment remodeling of the stroma and resistance to anti-cancer treatments [88,92,95]. For instance angiogenesis, another important cancer hallmark, is modulated after hypoxia-induced UPR activation by modulating expression of several proangiogenic mediators as such VEGF and angiogenic inhibitors including THBS1, CXCL14 and CXCL10 [88,138]. Moreover, UPR activation is also observed in tumor associated cells such as endothelial cells and infiltrating macrophages, lymphocytes and dendritic cells to increase their ability to support tumor growth by promoting neo-angiogenesis and by providing important secreted growth factors [90,93]. Currently, many studies are further documenting the involvement of ER stress in cancer hallmarks [9,10]. The following section of this review will be mainly focused on the links between UPR and tumor migration/invasion.

## 4. Connections between UPR Signaling and Tumor Cell Migration

Growing evidence suggests that the UPR is an important regulator of different steps of the tumor migration and metastasis. The three sensors of the UPR have been recently linked to tumor cell migration/invasion processes such as ECM and actin cytoskeleton remodeling and cytoskeleton reorganization, modification of cellular adhesion, activation of signaling pathways associated with cell mobility, and EMT [7,91,94,95].

### 4.1. Links between UPR Sensors Activation and Cancer Metastasis

The IRE1/XBP1 axis has been the most extensively correlated with cancer progression and metastasis. Importantly, studies with tumor samples from patients with colorectal carcinoma, breast cancer and oral squamous cell carcinoma, described the overexpression of IRE1 or XBP1 in metastatic samples compared to the primary tumors [139,140,141,142]. In addition, elevated levels of XBP1 at primary tumors are statistically associated to the presence of distant metastasis in patients with esophageal carcinoma, hepatocellular carcinoma and oral squamous cell carcinoma [143,144,145]. XBP1s overexpression at the primary tumor was correlated with intrahepatic invasion and distant metastasis in hepatocellular carcinoma [144]. In pancreatic cancers, latent liver metastases are developing from quiescent single disseminated cancer cells (DCCs) that evade to the anti-tumor immune response [146]. Intriguingly, these DCCs shut down IRE1 activity leading to escape from CD8 T cell cytotoxicity by down-regulating MHC class I molecules expression. Restoration of IRE1 signaling branch by overexpressing XBP1s in DCCs leads to the outgrowth of liver macro-metastatic lesions [146]. These findings suggest that IRE1 activation might be important for the initial and final steps in metastasis, like tumor cell dissemination and the formation of macro-metastasis, with a temporary downregulation for avoiding anti-tumor immune response. Besides IRE1, PERK activation has been also linked to tumor invasiveness. In triple negative breast cancers (TNBC), PERK activation characterized by a cancer-specific PERK signaling gene set is associated with distant metastasis [147]. Also, overexpression of ATF4, a component of PERK pathway, is associated with lymph node metastasis in esophageal squamous cell carcinoma [148]. In vivo experiments demonstrated that ATF4 induce cell invasion and metastasis stimulating MMP2 and MMP7 expression [148]. Although more accurate experiments are required, this evidence shows that UPR activation might be relevant for the development of metastatic lesions.

### 4.2. UPR-Dependent Control of ECM Protein Production and ECM Remodeling

ECM remodeling is an important step to allow tumor migration/invasion. ECM degradation is a key phenomenon in tumor cells migration through the adjacent tissues. In addition, the regulation of cell/ECM interactions determines the cell ability to migrate, i.e., strong cell adhesion to the ECM will limit cell invasion [30]. UPR-regulated molecules such as ECM components, i.e., collagens and fibronectin, enzymes that cleave ECM, i.e., cathepsin and MMPs and adhesion molecules, i.e., integrins, are involved in this process.

#### 4.2.1. ECM Remodeling by the IRE1/XBP1s Signaling Axis

One important process in metastasis is the invasion allowed by the degradation of the ECM through the expression of MMPs [149]. In esophageal squamous cell carcinomas, XBP1s overexpression promotes cell invasion and metastasis through the upregulation of MMP9, one of the MMPs most widely associated with cancer progression [143]. Similarly, XBP1 deficiency in oral squamous cell carcinoma cells impairs cell invasion and leads to a decrease in the expression of invasion-associated genes including MMP1, MMP3 and PLAUR [141]. Intriguingly, in GBM, IRE1 signaling is found to negatively modulate cell migration and invasion [137,150,151,152,153]. Gene expression profiling reveals that loss of enzymatic IRE1 activity results in an upregulation of ECM proteins, by negatively regulating the expression of SPARC through the RIDD-mediated degradation of its mRNA, a protein associated with changes in cell shape, synthesis of ECM and cell migration [151]. In addition, the expression of genes involved in cancer cell migration including ECM components (i.e., collagens), MMPs and chemokines is under the control of IRE1 activation in GBM cells [152].

#### 4.2.2. PERK-Dependent Regulation of MMPs in Cancers

As described above for IRE1, PERK is also found to contribute to ECM reorganization in cancer cells. For instance, in esophageal squamous carcinoma cells, ATF4 directly controls tumor migration in vitro and in vivo by regulating the expression of the metalloproteinases MMP2 and MMP7 that, in turn, facilitate this process via the ECM remodeling [148]. Interestingly, ATF4 has been described as a potential poor prognostic biomarker in this cancer type [148]. In chronic myeloid leukemia, eIF2α is constitutively phosphorylated and enhances invasive ability of tumor cells but also tumor associated stromal fibroblasts by modulating ECM remodeling through cathepsin and MMPs expression via the induction of ATF4 [154]. Interestingly, TRAM2 (for translocation associated membrane protein 2), a component of the SEC61 translocation channel located at ER, is highly expressed in oral squamous cell carcinoma and has a main role in metastasis by controlling PERK activation and the expression of MT1-MMP, MMP2, and MMP9 [155]. Breast cancer cell lines exhibit increased secretion of ECM proteins that perturbs ER morphology due to the overload in secretory proteins and show a constitutively activated PERK/eIF2α/ATF4 axis [156].

#### 4.2.3. ECM Remodeling upon ATF6 Activation

Little has been described so far on the potential role of ATF6 in modulating tumor cell migration/invasion. One recent study reports that ATF6 activation, upon ER stress induced by gemcitabine, leads to the increased expression of PLAU, a serine protease involved in the degradation of the ECM. Its activation is, in turn, associated with enhanced migration properties of pancreatic cancer stem cells [157]. Also, ADAM17, a member of the disintegrins and metalloproteases family that promotes tumor invasiveness and is found to be up-regulated in breast, gastric ovary and prostatic cancers and is induced by ATF6 in breast cancer cells [158]. Interestingly, PERK/eIF2α/ATF4 UPR arm also regulates ADAM17 expression as ATF4 binding sites are present in the ADAM17 promoter and PERK activation induces the ADAM17 protein release [158].

### 4.3. Involvement of the UPR-Dependent Secretome in Tumor Migration

Tumor cell migration depends on the interaction with the microenvironment, extracellular matrix adhesion, cell-cell contacts and matrix remodeling. Cytokines and growth factors that are secreted in the tumor microenvironment regulate all of these processes and therefore control the invasion capacity of tumor cells. These molecules can be secreted by both the tumor cells (autocrine signals) and by the surrounding non-tumor cells (paracrine signals), controlling the initial steps for the metastatic cascade and allowing tumor cell adaptation to environmental changes. The different UPR sensors have been involved in the production of pro-migratory cytokines and chemokines. IRE1 has been described to regulate the secretion of several factors that control tumor angiogenesis that can also affect tumor cell migration. For instance, in GBM, the inhibition of IRE1 decreases the expression of proangiogenic factors such as VEGFA, IL1β, IL6, and CXCL8 (also named IL8) and leads to a reduction of angiogenesis [150,151]. Moreover, IRE1 activity affects the adhesion, migration and invasion properties of GBM tumor cells [150,151,152] by controlling the production of the chemokines/cytokines IL6, CXCL8, and CXCL3, all involved in these processes [150,152]. Selective impairment of IRE1 RNase increase invasion, vessel co-option capacity and mesenchymal features in U87 glioma cells [153]. Interestingly, in colorectal cancers high XBP1s expression is associated with metastatic tumors in patients and with cancer cell invasion in vitro by controlling VEGFR2 expression [139]. In intestinal cancer cells, early growth response protein 1 (EGR1), an important transcription factor that controls the expression of chemokines/cytokines involved in tumor metastasis such as CCL2 and CXCL1, is positively regulated upon activation of PERK and ATF6 [159]. Suppression of PERK or targeting ATF6 decreased EGR1 expression levels as well as EGR1-associated chemokine expression. Interestingly, ATF3 through a direct interaction with histone deacetylase 1 (HDAC1) mediate EGR1 suppression [159]. PERK activation also increases VEGFA expression in medulloblastoma, which favors tumor migration through an autocrine manner by interacting with its receptor VEGFR2 [160]. In melanoma, both ATF6 and PERK branches of the UPR are involved in the induction of the fibroblast growth factors FGF1/2 increasing cancer cell migration in vitro [161].

### 4.4. UPR-Mediated Regulation of EMT in Cancers

In recent years, the EMT and UPR activation mainly through IRE1 and PERK signaling pathways have been closely linked to cancer progression in many models [7,91,94,156,162]. EMT-like phenotypes are induced upon UPR activation including cellular morphological changes and modulation of EMT markers, i.e., E-cadherin and vimentin [140,156,163]. Importantly, the common chemotherapeutic drugs used in cancers induce ER-stress mediated EMT, independent of the cancer type [164]. PERK activation is mandatory for tumor cells to invade and metastasize [147]. Furthermore, EMT gene expression signature has been correlated with ECM protein secretion and ATF4 expression (but not XBP1) in various cancers including breast and colon [156]. Inhibition of the PERK/eIF2α/ATF4 signaling axis with acriflavine (an antiseptic agent that also targets HIF1 pathway) prevents EMT at the cellular and molecular levels (i.e., no change in cellular morphology and no induction of EMT markers as such E-cadherin, vimentin, SNAI1, SPOCK1 and TWIST1); and inhibits the tumor cell migration (Figure 2, (2)) [165]. However, other studies indicate that XBP1s increases the metastatic potential of tumor cells by the induction of the expression of several EMT transcription factors, including SNAI1, SNAI2, ZEB2 and TCF3 [140,144,163,166]. The induction of these transcription factors for the IRE1/XBP1s signaling is dependent of lysyl oxidase-like 2 (LOXL2). Overexpression of LOXL2 induces its accumulation in the ER and its interaction with GRP78 inducing IRE1/XBP1s branch activation [163]. The inhibition of the RNase activity of IRE1 using small molecules reduced EMT markers expression patterns in breast cancer cells [140].

### 4.5. UPR-Dependent Regulation of Other Molecular Actors of Tumor Cell Migration

#### 4.5.1. Direct Interaction between IRE1 and Filamin A

We have recently uncovered a novel mechanism of cell migration regulation underlying IRE1 function. Using an interactome screening, FLNA is identified as a major IRE1-binding partner in non-cancer mouse and human cells [167]. FLNA is a 280 kDa actin crosslinking protein involved in the regulation of cytoskeleton remodeling through a direct phosphorylation at serine 2152 [168]. Remarkably, the regulation of cytoskeleton dynamics by IRE1 is independent of its canonical RNase activity, but instead IRE1 serves as a scaffold that recruits FLNA, scaffolding to PKCα, to increase FLNA phosphorylation. Using genetic manipulation, it was determined that deletion of IRE1 impaired actin cytoskeleton dynamics at the protruding and retracting areas. These finding were corroborated in zebra fish, drosophila and mouse models. In addition, using a panel of tumor cell lines, IRE1 silencing decreased tumor cell migration [151,169]. This discovery unveils the possibility of direct interaction between IRE1 and the cytoskeleton network which could also take place in cancer cells (Figure 1, (1) and (4)) [167].

#### 4.5.2. HIF1α Regulation by XBP1s

Basal XBP1s expression has been described in TNBCs and has a key role on tumorigenicity and tumor dissemination [170]. According to genome-wide mapping to determine XBP1s regulatory network, XBP1s interacts with HIF1α forming a transcriptional complex that enhances the expression of HIF1α-regulated genes by promoting the recruitment of RNA polymerase II [170]. It is well documented that the HIF1α transcriptional program plays a key role in critical steps of metastasis like EMT, extravasation and metastatic niche formation [171]. Furthermore, silencing of XBP1 decreased the formation of lung metastases in an orthotopic TNBC xenograft mouse model (Figure 2, (2)) [170].

#### 4.5.3. Dual Functions of CREB3L1 Induced by ER Stress on Tumor Migration

CREB3L1 (so called OASIS) is a transcription factor initially described in human astrocytes [172] and later considered as an ER stress sensor [173]. This protein is located at the ER membrane and under ER stress, CREB3L1, like ATF6, is exported to the Golgi apparatus and cleaved by S1P and S2P proteases. The membrane-free cytosolic domain is released and translocates to the nucleus to act as a transcription factor regulating the expression of several genes including ER chaperones such as GRP78, and CREB3L1 itself [173]. Using a bioinformatic approach that integrates gene mutations and DNA methylation changes, CREB3L1 was identified as an important regulatory driver in prostate cancer [174,175]. In glioma cells, ER stress induces CREB3L1 that, in turn, negatively modulates the expression of chondroitin sulfate proteoglycan and is associated with increased ability of tumor cell migration/invasion [176]. Surprisingly, CREB3L1 is lost in metastatic cells from breast and bladder tumors due to the methylation of its gene (in the promoter region and the first intronic region) leading to an epigenetic silencing [175,177]. Restoration of CREB3L1 expression in metastatic cells dramatically reduces their migration/invasion ability [175,177]. Importantly, CREB3L1 is transcriptionally regulated downstream of PERK via ATF4 induction but this also requires additional signaling molecules from the EMT pathway such as COL1A1, COL1A2, and FN1 [147]. Remarkably, CREB3L1 expression is a predictive marker for distant metastasis in the mesenchymal subtype of TNBCs [147]. CREB3L1 increases breast tumor migration capacities through ECM production and remodeling, i.e., COL1A2 and FN1. CREB3L1 inhibition also reduces FAK activation, an important kinase that regulates cell/ECM interaction via its impact on ECM (Figure 1, (2)) [147]. 

#### 4.5.4. LAMP3 Regulation by PERK Signaling in Cancers

Under hypoxic conditions, the PERK/ATF4 axis is activated and promotes breast tumor cell migration/invasion through the up-regulation and activation of LAMP3, a lysosomal-associated membrane protein [178]. PERK-mediated eIF2α phosphorylation also induces LAMP3-dependent cervix cancer cell migration under hypoxia [179]. Importantly, LAMP3 expression is also associated with metastasis and poor prognostic in breast, cervix and colorectal cancers and head and neck squamous carcinomas [178,179,180,181,182]. Although the LAMP family members are described as lysosomal membrane proteins, their cell surface expression is often observed in cancer cells. The biologic function of LAMP3 in tumor migration and metastasis needs therefore to be further characterized. As described with LAMP1, LAMP3 might participate to the membrane ruffles and filopodia in migrating tumor cells (Figure 1, (1)) [183].

## 5. Conclusions: UPR Signaling and Cell Migration as Future Targets in Cancer Therapy

Cancer cell migration/invasion has appeared as an important axis to target in the perspectives of anti-cancer therapies development [2,5]. As described above, tumor cell migration is linked to UPR signaling, thereby opening new therapeutic avenues. Interestingly, several inhibitors of the ER stress sensors have been reported to affect tumor migration. For instance, ATF6 inhibitors of the flavonoid family extracted from plants, i.e., apigenin, baicalein, kaempferol; display a strong effect on inhibiting tumor migration of the large range of cancer types including brain, breast, liver, lung, pancreas and skin, however this might be due to off target effects [184,185,186,187,188,189,190,191,192,193,194,195]. They mainly modulate MMP2 and MMP9 metalloproteinases expression [184,188,190,191,192], interfere with the EMT process through the regulation of SNAI1 and SLUG [187,189,194] and affect the AKT and MAPK signaling pathways [186,190,191,192,194,195]. Like flavonoid molecules, another ATF6 inhibitor melatonin modulates important kinases FAK, SRC and ROCK1 involved in tumor migration [195,196,197]. IRE1 inhibitors such as quercetin and sunitinib also inhibit tumor migration by modulating the same molecular actors of the ECM remodeling and intracellular signaling pathways, i.e., metalloproteinases and kinases, but again, these effects were not yet proven to occur through the inhibition of IRE1 [198,199,200,201]. More specific molecules that inhibit the PERK/eIF2α branch also affect tumor migration. The PERK inhibitor GSK2606414 blocks brain tumor cell migration [160], but this inhibitor is also known to target RIPK1 and c-KIT [202,203]. Subtoxic doses of eIF2α phosphatase GADD34/PP1c inhibitors guanabenz or salubrinal reduce breast and bone cancer cell migration/invasion through the reduction of SRC [204] and RAC1 [204,205] activity and through the modulation of MMP13 expression (for salubrinal) [204]. Altogether, although none of the UPR inhibitors are currently tested on clinical trials for cancer patients, these findings highlight the need for clarifying the molecular mechanisms occurring under UPR that control tumor migration/invasion. Better understanding of these mechanisms will allow to more specifically target the relevant actors to prevent tumor invasion and metastasis; and therefore, improve current therapeutic approaches for patients with cancer diseases.

## Figures and Tables

**Figure 1 cancers-11-00631-f001:**
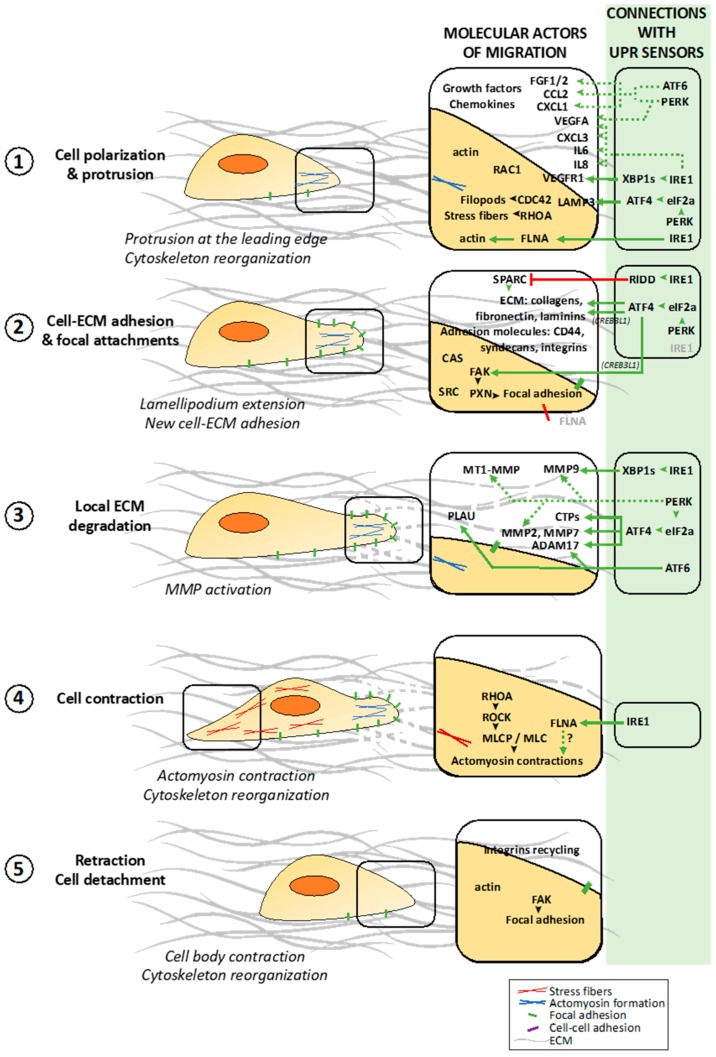
The cellular processes and molecular actors involved in cancer cell migration/invasion and their links to the Unfolded Protein Response (UPR) sensors. Cancer cells start to polarize via cytoskeleton reorganization at the leading edge (1) and generate new cell-matrix contacts (2). The proximal extracellular matrix (ECM) surrounding the leading edge is degraded by metalloproteinases (MMPs) activation to allow cell movement (3). Finally, cell contractions (4) and retractions allowed by cytoskeleton reorganizations, synchronized with cell-matrix detachments (5), lead the movement of the cell body. The molecular partners involved in the different cancer cell migration steps are presented in the associated boxes. The UPR sensors and their down-stream pathways that control the migration associated molecules are indicated in the green boxes (i.e., direct (solid lines) or indirect (dotted lines) links).

**Figure 2 cancers-11-00631-f002:**
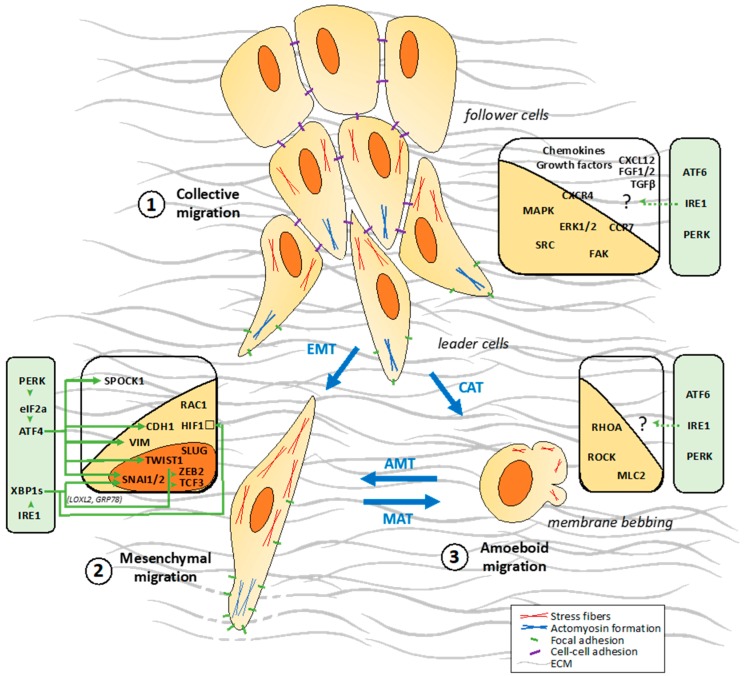
The migration modes used in cancer cell migration/invasion and their links to the UPR sensors. Cancer cells migrate either collectively (1) or individually according to the mesenchymal (2) or amoeboid (3) modes. The latter modes involve cell reprogramming processes including epithelial-to-mesenchymal (EMT), mesenchymal-to-amoeboid (MAT) and amoeboid-to-mesenchymal (AMT) transitions. Although less characterized, a collective-to-amoeboid transition (CAT) has been also documented. The molecules involved in these different cancer cell migration modes are presented in the associated boxes. The UPR sensors and their down-stream pathways that control the migration associated molecular partners are indicated in the green boxes.

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
