# Peer review of "Emerging Roles of the Endoplasmic Reticulum Associated Unfolded Protein Response in Cancer Cell Migration and Invasion"

_cancers, 2019, doi:10.3390/cancers11050631_

Round 1
Reviewer 1 Report
This is an extensive review covering detail molecular mechanisms of cancer migration and invasion and the role of ER-UPR in modulation of these processes. The manuscript is well written and the figures are very good. There are only few minor comments.
1. The title of manuscript is inappropriate as the entire review only focused on ER-UPR. There are three different kinds of UPRs, ER-UPR, mitochondrial UPR and cytosolic UPR. So, authors need to change the title slightly to reflect the review is specific on ER-UPR.
2. In Page 3, Figure 1 legend, line 65, “…molecular actors involved in cancer cell migration/invasion”. It should be migration only, not migration/invasion as described in page 2, line 59-60.
3. In page 9, ATF6, what genes are regulated by active ATF6 and the isoforms of ATF6 α and β should be mentioned.
4. In page 10, 3.1.4 PERK, role of NRF2 after phosphorylated by PERK should be mentioned.
5. In Page 10, line 350, spelling mistake, Sancer.
6. Potential triggers of ER stress in cancers need to be explored in 3.2. For example, hypoxia in solid tumour or other mechanisms such as mutation on ER-UPR regulators etc?
7. It will be clearer if authors add a diagram to show all 3 ER-UPR signalling pathways and only text description is a bit quite difficult to follow especially for those readers do not have much knowledge about UPR.
8. In Page 13, 4.4. UPR-Mediated Regulation of EMT in Cancers. Authors need to mention which UPR pathways in regulation of EMT are likely cancer’s specific.
9. In conclusion, it is inappropriate to have a subheading 5.1. It needs to be removed.
Author Response
We would like to thank you for the handling of our manuscript (cancers-492106) and the reviewers for their time and constructive critiques. We are pleased to submit a revised version of our work entitled “Emerging roles of the Endoplasmic Reticulum associated Unfolded Protein Response in cancer cell migration and invasion” by Limia and colleagues for publication as a review article in Cancers.
#Reviewer1
This is an extensive review covering detail molecular mechanisms of cancer migration and invasion and the role of ER-UPR in modulation of these processes. The manuscript is well written and the figures are very good. There are only few minor comments.
We thank the reviewer for this kind introduction.
1. The title of manuscript is inappropriate as the entire review only focused on ER-UPR. There are three different kinds of UPRs, ER-UPR, mitochondrial UPR and cytosolic UPR. So, authors need to change the title slightly to reflect the review is specific on ER-UPR.
The title has been changed as requested by the reviewer:
“Emerging roles of the Endoplasmic Reticulum associated Unfolded Protein Response in cancer cell migration and invasion.”
2. In Page 3, Figure 1 legend, line 65, “…molecular actors involved in cancer cell migration/invasion”. It should be migration only, not migration/invasion as described in page 2, line 59-60.
The Figure 1 legend has been changed consequently:
“Figure 1: The cellular processes and molecular actors involved in cancer cell migration and their links to the UPR sensors.”
3. In page 9, ATF6, what genes are regulated by active ATF6 and the isoforms of ATF6 α and β should be mentioned.
The genes targeted by active ATF6 and a description of the ATF6 isoforms are now provided in page 9 paragraph 3.1.2:
“ATF6 is a ER-localized protein that exists in two isoforms α and ß forming homo- and heterodimers (92, 93). Compared to the isoform ATF6ß, ATF6α appears to be a very potent transcription factor (94).”
“…induces the transcription of genes mainly involved in protein folding and ERAD such as calreticulin, GRP78, HERPUD1 and SEL1L (7, 78, 101, 102).”
4. In page 10, 3.1.4 PERK, role of NRF2 after phosphorylated by PERK should be mentioned.
This point is now indicated in part 3.1.4 of the revised version of the manuscript:
“Phosphorylation of cytoplasmic NRF2 leads to its dissociation from KEAP1 and its nuclear import (109).”
5. In Page 10, line 350, spelling mistake, Sancer.
This typo has been corrected.
“3.2. Roles of the ER stress sensors in cancer”
6. Potential triggers of ER stress in cancers need to be explored in 3.2. For example, hypoxia in solid tumour or other mechanisms such as mutation on ER-UPR regulators etc?
More details about this point are now provided in part 3.2:
“During tumor development, cancer cells are exposed to several challenges such as acute demands of protein synthesis due to oncogenes expression and drastic extracellular conditions linked to hypoxia or low nutrient availability. These challenges require efficient UPR allowing the cancer cells to cope and adapt”
7. It will be clearer if authors add a diagram to show all 3 ER-UPR signalling pathways and only text description is a bit quite difficult to follow especially for those readers do not have much knowledge about UPR.
We thank the reviewer for this comment. We understand that part 3.1 is difficult to follow for the readers that have no expertise in the UPR field. However, we did not want to duplicate what is largely and fully described in other reviews on the UPR. Instead we did just want to give a brief overview of the different ER UPR signaling pathways. To answer to the reviewer’s comment, we have added a sentence in the revised version of the manuscript to underline this point (see paragraph 3.1):
“Herein, we will provide a brief overview of the UPR signaling pathways and invite the readers to refer to the following reviews on UPR for more details (6-8, 75-85).”
8. In Page 13, 4.4. UPR-Mediated Regulation of EMT in Cancers. Authors need to mention which UPR pathways in regulation of EMT are likely cancer’s specific.
This point is now provided in part 4.4 of the revised version of the manuscript:
“In recent years, the EMT and UPR activation mainly through IRE1 and PERK signaling pathways have been closely linked to cancer progression in many models (7, 80, 83, 144, 150).”
9. In conclusion, it is inappropriate to have a subheading 5.1. It needs to be removed.
We thank the reviewer for underlining this mistake. This has been corrected in the revised version of the manuscript.
Reviewer 2 Report
The manuscript titled “Emerging Roles of the Unfolded Protein Response in Cancer Cell Migration and Invasion” by Limia CM et al is a very well written review article. This timely review article addresses a very important aspect of research. UPR reprograms protein output and allows the cell to cope with the stress condition. However, cancer cells are under chronic stress and UPR has a significant role in cancer progression. This manuscript can be improved as follows.
Collective migration is well described, however, the mesenchymal migration should be better described with the molecular pathways/signatures.
STAT3 is implicated in mesenchymal GBM progression via SNAIL and TWIST. This should be described in details.
Angiogenesis is implicated in the progression of various cancers such as HNSCC, GBM, and breast cancer. There is a limited discussion on this point. The authors should expand, if possible in a dedicated section, these topics. Particularly, the role of EGFR-HIF1alpha axis, as well as VEGF should be described in details.
Line # 346 should read as “GADD34 is activated”
Line # 350 “Sancer” should be “Cancer”
Line # 367-368 “ This review……” this statement is not appropriate here.
Author Response
We would like to thank you for the handling of our manuscript (cancers-492106) and the reviewers for their time and constructive critiques. We are pleased to submit a revised version of our work entitled “Emerging roles of the Endoplasmic Reticulum associated Unfolded Protein Response in cancer cell migration and invasion” by Limia and colleagues for publication as a review article in Cancers.
#Reviewer2
The manuscript titled “Emerging Roles of the Unfolded Protein Response in Cancer Cell Migration and Invasion” by Limia CM et al is a very well written review article. This timely review article addresses a very important aspect of research. UPR reprograms protein output and allows the cell to cope with the stress condition. However, cancer cells are under chronic stress and UPR has a significant role in cancer progression. This manuscript can be improved as follows.
We thank the reviewer for the comments made and we have improved the revised version of the manuscript as suggested by the reviewer.
Collective migration is well described, however, the mesenchymal migration should be better described with the molecular pathways/signatures.
We have improved the revised manuscript by including the following sentence in part 2.2.2:
“RAC1 is activated by DOCK3 (for Dedicated Of Cytokinesis 3) and its adaptor molecule NEDD9 to drive mesenchymal migration (34). Active RAC1 negatively regulates RHO/ROCK signaling inhibiting cell rounding and promoting mesenchymal movement. The mesenchymal mode requires ECM proteolysis to allow RAC1-dependent actin protrusions to pass through the ECM mesh (34). Integrins, membrane-type MMPs and other proteases co-localize at fiber binding sites to contribute ECM degradation. The activation of MMPs and uPA (for urokinase-type plasminogen activator) is required for maintenance of the phenotype and mesenchymal migration (11).”
STAT3 is implicated in mesenchymal GBM progression via SNAIL and TWIST. This should be described in details.
We thank the reviewer for the comment. We have improved the revised manuscript by including the following sentence in part 2.3.2:
“The IL6/STAT3 signaling axis is also involved in GBM cell migration by modulating the expression of metalloproteinases MMP2 and MMP9 (63, 64); as well as GRP1 (for Glioma Pathogenesis-Related 1) that contributes to GBM stem cell migration (65). STAT3 is also implicated in mesenchymal GBM progression by modulating the mucin-type protein podoplanin and the EMT-related transcription factors SNAIL and TWIST (66)”.
Angiogenesis is implicated in the progression of various cancers such as HNSCC, GBM, and breast cancer. There is a limited discussion on this point. The authors should expand, if possible in a dedicated section, these topics. Particularly, the role of EGFR-HIF1alpha axis, as well as VEGF should be described in details.
As we indicated in the abstract, the main focus of this review is to link UPR and tumor migration/invasion. As the reviewer underlined, angiogenesis is also an important factor in tumor progression but we believe that the point is off topic in the present review. However, we have added the following sentence in part 3.2. of the revised manuscript:
“For instance angiogenesis, another important cancer hallmark, is modulated after hypoxia-induced UPR activation by modulating expression of several proangiogenic mediators as such VEGF and angiogenic inhibitors including THBS1, CXCL14 and CXCL10 (82, 132).”
Line # 346 should read as “GADD34 is activated”
This is now corrected in part 3.1.4:
“GADD34 is activated downstream of CHOP”
Line # 350 “Sancer” should be “Cancer”
This typo has been corrected.
“3.2. Roles of the ER stress sensors in cancer”
Line # 367-368 “ This review……” this statement is not appropriate here.
We have modified the sentence in part 3.2. as followed:
“The following section of this review will be mainly focused on the links between UPR and tumor migration/invasion.”
Reviewer 3 Report
Cancer metastasis were through the cells migration/invasion to drive cancer progression leading to tumor expansion. Metastasis also largely contribute to most of the curative failures in cancers and in cancer-related mortality. Recently, targeting tumor invasion/migration has become a potential for the development of new types of anti-cancer treatments. In this review, the authors described the cellular and molecular aspects of the cell migration and their relevance to cancer cell migration/invasion, mainly focusing on brain and skin tumors, and how this can be connected to UPR signaling. Some aspects of the work, however, remain to be clarified.
comments:
The authors collected, classified and connected many relative results from references, and tried to demonstrate and explain the metastatic mechanism of UPR-related signaling in cancer cells.
The authors described the “FAK causes focal contact disassembly by yet unknown” in line 126-127 of manuscript. For example, two paper recently have showed Akt (https://www.ncbi.nlm.nih.gov/pubmed/23264741) and Dab2 (Disabled-2; https://www.ncbi.nlm.nih.gov/pubmed/19951918 ), respectively involved in the action of FAK caused focal contact for cell motility. The authors are advised to up-date the data to further classify it.
The authors explained “In contrast to the mesenchymal mode of invasion, in the amoeboid-like migration,……promoted by the RHO/ROCK signaling pathway” in line 126-127 of manuscript. They indicated the separated pathway for Rac-related and RHO/ROCK-related models of invasion. The authors may provide more information to further classify the correlation Rac and RHO/ROCK in mesenchymal mode of invasion and amoeboid-like migration.
The authors may provide some information of UPR inhibitors that were currently used in clinical or in clinical trial in this manuscript.
Author Response
We would like to thank you for the handling of our manuscript (cancers-492106) and the reviewers for their time and constructive critiques. We are pleased to submit a revised version of our work entitled “Emerging roles of the Endoplasmic Reticulum associated Unfolded Protein Response in cancer cell migration and invasion” by Limia and colleagues for publication as a review article in Cancers.
#Reviewer3
Cancer metastasis were through the cells migration/invasion to drive cancer progression leading to tumor expansion. Metastasis also largely contribute to most of the curative failures in cancers and in cancer-related mortality. Recently, targeting tumor invasion/migration has become a potential for the development of new types of anti-cancer treatments. In this review, the authors described the cellular and molecular aspects of the cell migration and their relevance to cancer cell migration/invasion, mainly focusing on brain and skin tumors, and how this can be connected to UPR signaling. Some aspects of the work, however, remain to be clarified.
We thank the reviewer for the comments made and we have improved the revised version of the manuscript consequently.
comments:
The authors collected, classified and connected many relative results from references, and tried to demonstrate and explain the metastatic mechanism of UPR-related signaling in cancer cells.
The authors described the “FAK causes focal contact disassembly by yet unknown” in line 126-127 of manuscript. For example, two paper recently have showed Akt (https://www.ncbi.nlm.nih.gov/pubmed/23264741) and Dab2 (Disabled-2; https://www.ncbi.nlm.nih.gov/pubmed/19951918 ), respectively involved in the action of FAK caused focal contact for cell motility. The authors are advised to up-date the data to further classify it.
We thank the reviewer for this comment. We have improved consequently the revised manuscript by adding the following sentences in part 2.1.3:
“FAK causes focal contact disassembly once phosphorylated by AKT1 (27).”
“Integrin endocytosis is mediated by clathrin, and its adaptor molecules ARH (for autosomal recessive hypercholesteremia) and DAG2 (for Disabled-2) (28)”
The authors explained “In contrast to the mesenchymal mode of invasion, in the amoeboid-like migration,……promoted by the RHO/ROCK signaling pathway” in line 126-127 of manuscript. They indicated the separated pathway for Rac-related and RHO/ROCK-related models of invasion. The authors may provide more information to further classify the correlation Rac and RHO/ROCK in mesenchymal mode of invasion and amoeboid-like migration.
We thank the reviewer for this comment. We have also included the following sentence in part 2.2.3 of the revised manuscript by adding:
“This underlines a mutual antagonistic role between RHO-family GTPases RAC1 and RHO/ROCK that display exclusive functions in mesenchymal and amoeboid-like migration respectively (38-40). By modulating these RHO GTPases, several molecules control the mesenchymal/amoeboid switch as such FILGAPP and ARHGAP22 in cancer cells (34, 40, 41).”
The authors may provide some information of UPR inhibitors that were currently used in clinical or in clinical trial in this manuscript.
To our acknowledge, none of the UPR inhibitors are currently tested in clinical trials with cancer patients. The following sentence has been included in part 5 of the revised version of the manuscript:
“..although none of the UPR inhibitors are currently tested on clinical trials for cancer patients, …”
Reviewer 4 Report
Here, the authors provide a nice, comprehensive and focused overview on the emerging role of the so call unfolded protein response (UPR) on malignant cell transformation and tumor progression. They first provide a brief overview of ER stress signaling and introduce the mechanism and the molecular actors involved in cancer cell migration and invasion. Subsequently they elegantly describe how these actors and machismos can be connected to UPR signaling. Overall this is a nice conceptual review on an emerging and important field of research.
Major comment:
Lane 150ff: Whereas CXCR4 signaling directly controls cell migration, the atypical chemokine receptor CXCR7 (which has been renamed to ACKR3! See PMID 24218476) scavenges CXCL12 to form chemokine gradients, but does not signal to guide cell migration. However, CCR7 contributes to (collective) cancer cell dissemination, migration and invasion (eg. PMID: 25019368). Please also revise and correct the information in Figure 2.
Minor comment:
Lane 129/130: the sentence should probably read as ‘All these steps occur sequentially and polarized…’
Lane 333: correct k in NFkB to Greek letter.
Lane 350: correct Sancer to Cancer
Lane 454: correct IL8 to CXCL8
Author Response
We would like to thank you for the handling of our manuscript (cancers-492106) and the reviewers for their time and constructive critiques. We are pleased to submit a revised version of our work entitled “Emerging roles of the Endoplasmic Reticulum associated Unfolded Protein Response in cancer cell migration and invasion” by Limia and colleagues for publication as a review article in Cancers.
#Reviewer4
Here, the authors provide a nice, comprehensive and focused overview on the emerging role of the so call unfolded protein response (UPR) on malignant cell transformation and tumor progression. They first provide a brief overview of ER stress signaling and introduce the mechanism and the molecular actors involved in cancer cell migration and invasion. Subsequently they elegantly describe how these actors and machismos can be connected to UPR signaling. Overall this is a nice conceptual review on an emerging and important field of research.
We thank the reviewer for this kind introduction.
Major comment:
Lane 150ff: Whereas CXCR4 signaling directly controls cell migration, the atypical chemokine receptor CXCR7 (which has been renamed to ACKR3! See PMID 24218476) scavenges CXCL12 to form chemokine gradients, but does not signal to guide cell migration. However, CCR7 contributes to (collective) cancer cell dissemination, migration and invasion (eg. PMID: 25019368). Please also revise and correct the information in Figure 2.
We thank the reviewer for this comment. We have modified the text in part 2.2.1 and the figure 2 consequently.
Minor comment:
Lane 129/130: the sentence should probably read as ‘All these steps occur sequentially and polarized…’
We thank the reviewer for this comment. The sentence in part 2.1.3 is now corrected:
“All these steps are sequential and polarized through the cell”
Lane 333: correct k in NFkB to Greek letter.
This is corrected in part 3.1.3.
Lane 350: correct Sancer to Cancer
This typo has been corrected:
“3.2. Roles of the ER stress sensors in cancer”
Lane 454: correct IL8 to CXCL8
As requested by the reviewer, IL8 is now corrected to CXCL8 in part 4.3 of the revised manuscript:
“…such as VEGFA, IL1β, IL6, and CXCL8 (also named IL8)…”
and “…by controlling the production of the chemokines/cytokines IL6, CXCL8, and CXCL3…”